# Does the Combined Effect of Resistance Training with EPO and Iron Sulfate Improve Iron Metabolism in Older Individuals with End-Stage Renal Disease?

**DOI:** 10.3390/nu13093250

**Published:** 2021-09-18

**Authors:** Hugo de Luca Corrêa, Víctor Manuel Alfaro-Magallanes, Sting Ray Gouveia Moura, Rodrigo Vanerson Passos Neves, Lysleine Alves Deus, Fernando Sousa Honorato, Victor Lopes Silva, Artur Temizio Oppelt Raab, Beatriz Carneiro Habbema Maia, Isabela Akaishi Padula, Lucas Santos de Gusmão Alves, Rafaela Araújo Machado, Andrea Lucena Reis, Jonato Prestes, Carlos Ernesto Santos Ferreira, Luiz Sinésio da Silva Neto, Fernanda Silveira Tavares, Rosângela Vieira Andrade, Thiago dos Santos Rosa

**Affiliations:** 1Graduate Program of Physical Education, Catholic University of Brasilia, Federal District, Brasilia 71966-700, Brazil; stingraygm@hotmail.com (S.R.G.M.); rpassosneves@yahoo.com.br (R.V.P.N.); lys.deus@gmail.com (L.A.D.); ferhon@gmail.com (F.S.H.); victornutri@yahoo.com.br (V.L.S.); andrealucereis@gmail.com (A.L.R.); jonatop@gmail.com (J.P.); ernestobsb@gmail.com (C.E.S.F.); thiagoacsdkp@yahoo.com.br (T.d.S.R.); 2LFE Research Group, Department of Health and Human Performance, Faculty of Physical Activity and Sport Sciences (INEF), Universidad Politécnica de Madrid (UPM), 28040 Madrid, Spain; vm.alfaro@upm.es; 3Department of Medicine, Catholic University of Brasilia, Federal District, Brasilia 71966-700, Brazil; artur.t.oppelt@gmail.com (A.T.O.R.); beatriz.maia@a.ucb.br (B.C.H.M.); bela.akaishi@gmail.com (I.A.P.); lucasgusmao1204@gmail.com (L.S.d.G.A.); rafaelamacha99@gmail.com (R.A.M.); fernanda.endocrino@gmail.com (F.S.T.); 4Faculty of Medicine, Federal University of Tocantins, Para 77402-970, Brazil; luizneto@mail.uft.edu.br; 5Graduate Program of Genomic Sciences and Biotechnology, Catholic University of Brasilia, Federal District, Brasilia 71966-700, Brazil; rosangelavand@gmail.com

**Keywords:** iron metabolism, chronic kidney disease, hepcidin, erythropoietin, exercise training, anemia, nephrology

## Abstract

We sought to investigate the effects of resistance training (RT) combined with erythropoietin (EPO) and iron sulfate on the hemoglobin, hepcidin, ferritin, iron status, and inflammatory profile in older individuals with end-stage renal disease (ESRD). ESRD patients (*n*: 157; age: 66.8 ± 3.6; body mass: 73 ± 15; body mass index: 27 ± 3), were assigned to control (CTL; *n*: 76) and exercise groups (RT; *n*: 81). The CTL group was divided according to the iron treatment received: without iron treatment (CTL—none; *n* = 19), treated only with iron sulfate or EPO (CTL—EPO or IRON; *n* = 19), and treated with both iron sulfate and EPO (CTL—EPO + IRON; *n* = 76). The RT group followed the same pattern: (RT—none; *n* = 20), (RT—EPO or IRON; *n* = 18), and (RT—EPO + IRON; *n* = 86). RT consisted of 24 weeks/3 days per week at moderate intensity of full-body resistance exercises prior to the hemodialysis section. The RT group, regardless of the iron treatment, improved iron metabolism in older individuals with ESRD. These results provide some clues on the effects of RT and its combination with EPO and iron sulfate in this population, highlighting RT as an important coadjutant in ESRD-iron deficiency.

## 1. Introduction

End-stage renal disease (ESRD) patients often present impaired intestinal absorption of dietary iron, blood losses, and low-grade chronic inflammation, which can lead to difficulties in achieving adequate iron status [1,2]. This adverse condition is one of the main causes of hyporesponsiveness to therapy involving erythropoiesis-stimulating agents (ESA), e.g., erythropoietin [2,3]. However, this supplementation, besides being associated with gastrointestinal discomfort, may impair the absorption of other medication, and alter the gut microbiota and systemic metabolome [2,3,4]. Thus, in the majority of cases, the patient requires an intravenous infusion of iron to treat this condition. Indeed, parenteral administration of iron is safer and more efficient than oral therapy, but not free from adverse events [4]. In this regard, patients with ESRD require oral iron supplementation to manage hemoglobin levels [1,2,3,5].

This adverse scenario motivates physicians and researchers to seek alternative and non-pharmacological treatments to counteract the adverse events related to ESA therapy and iron supplementation. A recent study from our laboratory outlined the positive effects of resistance training (RT) on iron deficiency in ESRD patients, suggesting this training model as an adjunct treatment for anemia [6]. In our study, 72.61% of the patients underwent EPO treatment (control group (CTL) = 69.73%; resistance training group (RT) = 75.30%) and 54.14% underwent oral iron supplementation (CTL = 55.26%; RT = 53.08%). Given that both treatments may influence multifactorial pathways related to inflammation and iron status, and RT is an important regulator of both aforementioned factors [6,7,8,9], two fundamental questions are raised here: (I) Does RT enhance the treatment with EPO and iron sulfate in ESRD older individuals? (II) Is RT an effective treatment on its own to improve iron status?

Attempting to answer this question, we sought to investigate the effects of RT combined with EPO and iron sulfate on the hemoglobin, hepcidin, ferritin, iron status, and inflammatory profile in older individuals with ESRD. We hypothesized that RT would enhance the effects of EPO and iron sulfate in this population. If confirmed, such findings might point to RT as an important adjunct therapy for iron deficiency that may work together with therapy involving erythropoiesis-stimulating agents.

## 2. Materials and Methods

### 2.1. Procedures

This study is part of a large trial [6]. Briefly, all participants involved in the study read and agreed with the written informed consent. The experimental protocols were approved by the Local Ethics Committee. All procedures were carried out conforming to the principles outlined in the declaration of Helsinki (1975). The participants were randomized into two groups by simple randomization: the control group (CTL; *n* = 76) and the resistance training group (RT; *n* = 81).

Here, we stratified the patients into six subgroups according to the iron treatment received. The CTL group was divided as follows: without iron treatment (CTL—none; *n* = 19), treated only with iron sulfate or EPO (CTL—EPO or IRON; *n* = 19), and treated with both iron sulfate and EPO (CTL—EPO + IRON; *n* = 38). The RT groups followed the same pattern (RT—none; *n* = 20), (RT—EPO or IRON; *n* = 18), (RT—EPO + IRON; *n* = 43), as described in Figure 1. The protocol of RT occurred before the hemodialysis section and was described in detail by Moura et al. [6].

### 2.2. Iron and ESA Treatments

The protocols of iron and ESA agents were performed according to the parameters established by the Ministry of Health in Brazil for ESRD [10]. Briefly, patients on hemodialysis are initially treated with one of the following options, later adjusted according to the therapeutic response: −50–100 UI/Kg, subcutaneously, divided into 1 to 3 applications per week; 50–100 UI/Kg, intravenously, divided into 3 applications per week. If after four weeks of treatment, the hemoglobin elevation is less than 0.3 g/dL per week: increase the dose by 25%, respecting the maximum dose limit, which is 300 IU/Kg/week per route subcutaneously and 450 IU/kg/week intravenously. If, after four weeks of treatment, the hemoglobin elevation is in the range of 0.3–0.5 g/dL per week: keep the dose in use. If, after four weeks, the hemoglobin elevation is greater than 0.5 g/dL per week or the hemoglobin level is between 12 and 13 g/dL: reduce the dose by 25% to 50%, respecting the minimum dose limit recommended, which is 50 UI/Kg/week subcutaneously. Temporarily suspend treatment if the hemoglobin level is above 13 g/dL.

Treatment should be continuous, targeting the hemoglobin at 11 g/dL. Temporary interruption of treatment is recommended if the hemoglobin level is above 13 g/dL, with resumption when hemoglobin levels are below 11 g/dL. Discontinuation should be considered in the event of a serious adverse event.

Iron III hydroxide saccharate is for intravenous use and is presented in 5 ml ampoules containing 100 mg of iron III (20 mg/mL). It must be diluted in 100 mL of saline solution and infused within 15 min, according to the manufacturer. A study demonstrates its safe use in shorter administration times, up to 5 min, without increasing adverse reactions.

### 2.3. Biochemical Analysis

Venous blood samples were obtained at baseline, and after 24 weeks of training, to measure the iron metabolism status and inflammatory profile. All analyses were described elsewhere [6].

### 2.4. Statistical Analysis

Initially, normality and homogeneity of data were verified using the Shapiro–Wilk and Levene tests, respectively. A three-way ANOVA 2 × 2 × 3 (Group × Time × iron treatment) was performed to compare groups. Deltas (post-pre) were obtained from all groups and compared using a two-way mixed ANOVA 2 × 3 (group × iron treatment) followed by Tukey’s post-hoc. Results were considered significant at *p* < 0.05. All statistical analyses were performed using SPSS Statistics for Windows, version 22.0 (released 2013) (IBM Corp., Armonk, NY, USA), and GraphPad Prism for Windows, version 8.0.0 (GraphPad Software, San Diego, CA, USA).

## 3. Results

Subjects that completed the RT protocol did not present adverse effects. Baseline characteristics are displayed in Table 1.

Patients from the CTL group did not display changes in iron, hemoglobin, ferritin, and hepcidin after this six-month assessment in all iron treatment groups (*p* > 0.05). RT increased hemoglobin from baseline in all groups. Only the RT group treated with EPO + IRON presented higher hemoglobin in relation to the CTL group treated with EPO + IRON. Serum iron increased in relation to baseline and the CTL groups. Ferritin only decreased from baseline. The RT group presented a decrease in hepcidin in relation to baseline and in relation to the CTL group that received the same iron therapy, as described in Figure 2.

Patients from the RT groups displayed an improvement in the inflammatory profile, presenting a decrease in TNF and IL6 and an increase in IL10 when compared to baseline and to the CTL groups, as described in Figure 3.

As displayed in Figure 4, regardless of the treatment, the RT group showed modulated hemoglobin, iron, ferritin, hepcidin, and cytokine levels. However, RT—IRON + EPO presented a lower decrease in hepcidin in relation to RT—none.

## 4. Discussion

The aim of the present study was to investigate the effects of RT combined with EPO and iron sulfate on the hemoglobin, hepcidin, ferritin, iron status, and inflammatory profile in older individuals with ESRD. Here, we found that regardless of iron treatment, RT appeared to improve serum iron homeostasis parameters in older subjects with chronic kidney disease. These results may point to two important insights: (1) that RT is an effective treatment for improving serum iron parameters in elderly patients with ESRD and (2) that conventional use of ESA + iron in this population is more effective in combination with RT. Interestingly, as RT alone improved serum iron availability, it may provide the opportunity to remove ferrous sulfate and its associated disturbances from the treatment in this population. This would limit iron accumulation and, in theory, further reduce hepcidin, as suggested by the greater reduction in hepcidin in the RT—none group compared to the RT—IRON + EPO group. Furthermore, although the small sample size did not allow us to perform this analysis, the combination of RT + ESA is probably the most promising treatment for lowering hepcidin levels. This combination would reduce the activation of the two main pathways of hepcidin synthesis: the JAK/STAT3 pathway, by lowering inflammation (RT effects), and the BMP/SMAD pathway, by increasing erythropoiesis activity and sequestering erythroferrone BMP2/6 ligands [11,12].

According to the results found by Agarwal et al. [5], non-hemodialysis CKD patients on oral iron therapy have improved hemoglobin, TIBC, transferrin saturation and ferritin. However, although our participants received oral iron supplementation, they were also on hemodialysis treatment, which is known to increase iron losses. It was estimated that CKD patients on hemodialysis lose 1 to 3 g of iron per year, which, coupled with the fact that hemodialysis patients have particularly impaired dietary iron absorption, appears to make oral iron supplementation poorly effective in improving iron markers [13]. In fact, oral iron supplementation was no better than placebo in improving anemia, improving or preventing iron deficiency, or reducing ESA dosage in hemodialysis patients [13,14,15,16]. Therefore, the findings from the present study can provide clues on the application of RT in this population to counteract iron-related diseases. To date, no studies have investigated the pooled effects of exercise plus ESA therapy or iron supplementation after 24 weeks.

Anemia and iron deficiency is a common complication of hemodialysis patients [1,2,3,5]. Because of that, ESA and iron supplementation may be considered to improve health-related parameters in this population [1,2]. RT has appeared as a non-pharmacological therapy to improve iron metabolism in this population [6,8,17]. The possibility of using RT as part of the treatment of iron deficiency may lead to relevant management of anemia biomarkers (Figure 2 and Figure 3). Moreover, it could lead to a cost reduction in the treatment of anemia in ESRD patients, due to the possibility of reducing or removing drugs administered for this purpose. Nonetheless, although this study may point to important insights for future research, the present findings should be interpreted cautiously, because we did not control and randomize the sample according to the iron treatment.

Anemia leads to reduced quality of life, fatigue, dyspnea, and impaired cognitive capacity. Furthermore, it is associated with a greater risk of adverse cardiovascular events and mortality. Therefore, these conditions usually end up leading to a greater number of hospitalizations, with increased costs for the health system. In Brazil, it is estimated that 133,464 patients were on dialysis in 2018, and the use of EPO was part of the treatment of more than 80% of these patients. The annual cost of treatment for anemia in patients with chronic kidney disease can reach US $3241.65 dollars, while the minimum monthly wage for Brazilian citizens is around US $222 dollars, leading to an economic burden for the treatment of anemia [18]. Therefore, a key finding of the present study was the improvement in iron, hemoglobin, ferritin and hepcidin markers, regardless of the use of EPO, suggesting that the application of RT in hemodialysis clinics would be more cost-effective.

The present manuscript had an important limitation: the study did not control for the time of iron supplementation or EPO, which might influence the dependent variables. We recommend further studies to control for this condition. As this study is an additional analysis of a larger trial [6], it was not initially designed for this subgroup analysis, which is why there is a low sample size for each iron treatment group. The lack of analysis related to nutritional markers and electrolytes also limits our study since it could influence iron metabolism. However, to date, this is the first study to demonstrate that RT is capable of inducing changes in iron metabolism regardless of iron treatment. Moreover, these additional findings may open perspectives about the combined effect of RT and pharmacological iron treatments and encourage further studies to be designed to answer this question in ESRD patients with iron deficiency.

## 5. Conclusions

We conclude that regardless of the iron treatment, RT could improve hemoglobin, iron, ferritin and hepcidin in older individuals with ESRD. These novel findings provide some clues on the combined effect of RT plus EPO and iron sulfate in this population. Moreover, RT alone may also be an effective strategy to improve iron metabolism in hemodialysis patients. Therefore, it is rational to infer that the application of RT programs should be strongly recommended in dialysis care. This would improve the prognosis of several ESRD patients, especially those with iron deficiency and anemia. Further studies are needed to determine whether the treatment with EPO and iron sulfate is more effective in treating iron deficiency and anemia when combined with exercise training in patients with ESRD.

## Figures and Tables

**Figure 1 nutrients-13-03250-f001:**
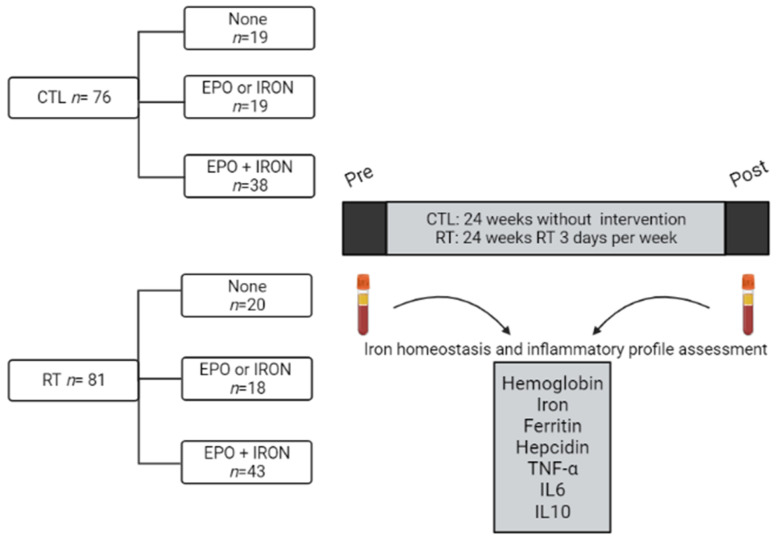
Experimental design. CTL: control group; RT: resistance training group; EPO: Erythropoietin; TNF: tumor necrosis factor; IL: interleukin.

**Figure 2 nutrients-13-03250-f002:**
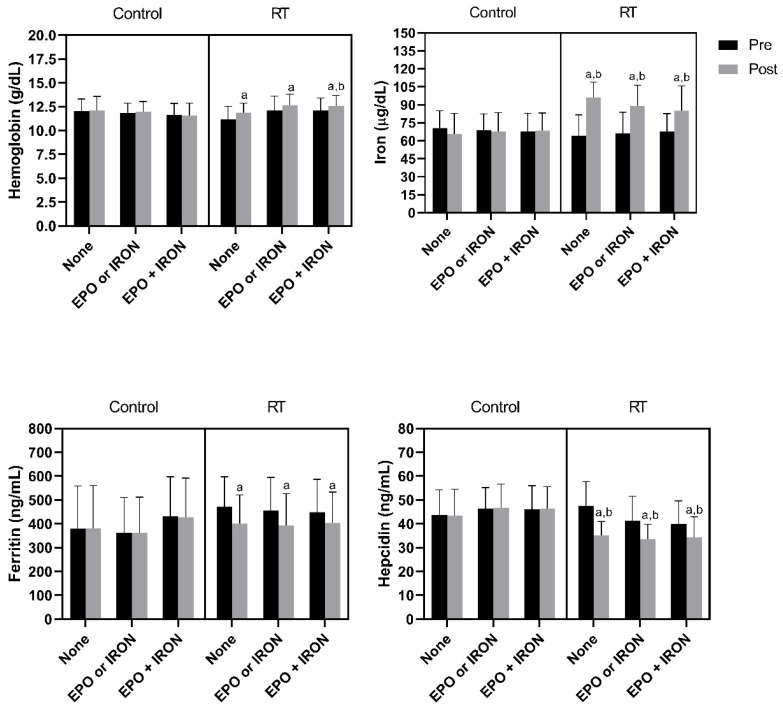
RT modulated iron, hemoglobin, ferritin and hepcidin levels in hemodialysis patients regardless of iron treatment. RT: resistance training; EPO: erythropoietin. Data expressed by mean ± SD. ^a^
*p* < 0.05 in relation to pre (within group and treatment). ^b^
*p* < 0.05 in relation to CTL post (between group and within treatment).

**Figure 3 nutrients-13-03250-f003:**
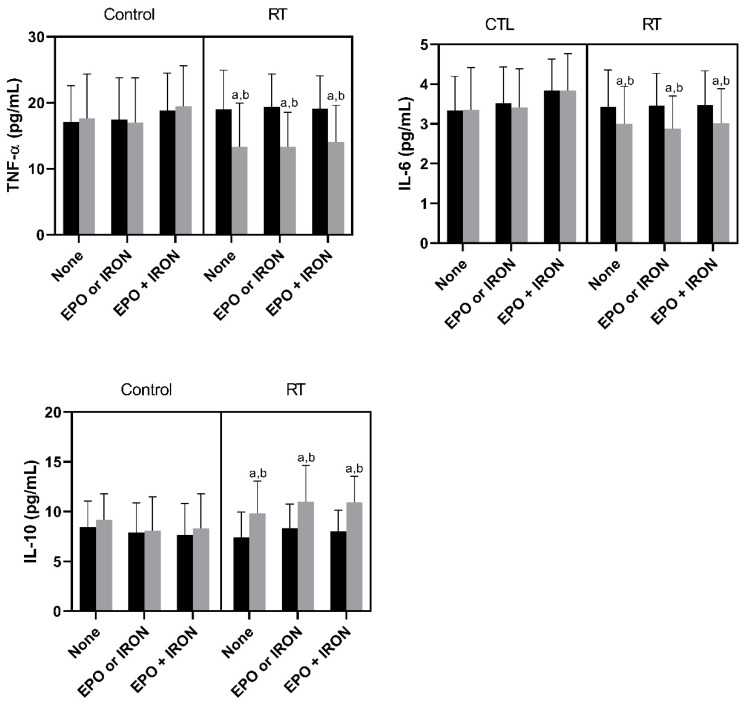
Cytokine modulation following RT in hemodialysis patients. RT: resistance training; EPO: erythropoietin; TNF: tumor necrosis factor; IL: interleukin. Data expressed by mean ± SD. ^a^
*p* < 0.05 in relation to pre (within group and treatment). ^b^
*p* < 0.05 in relation to CTL post (between group and within treatment).

**Figure 4 nutrients-13-03250-f004:**
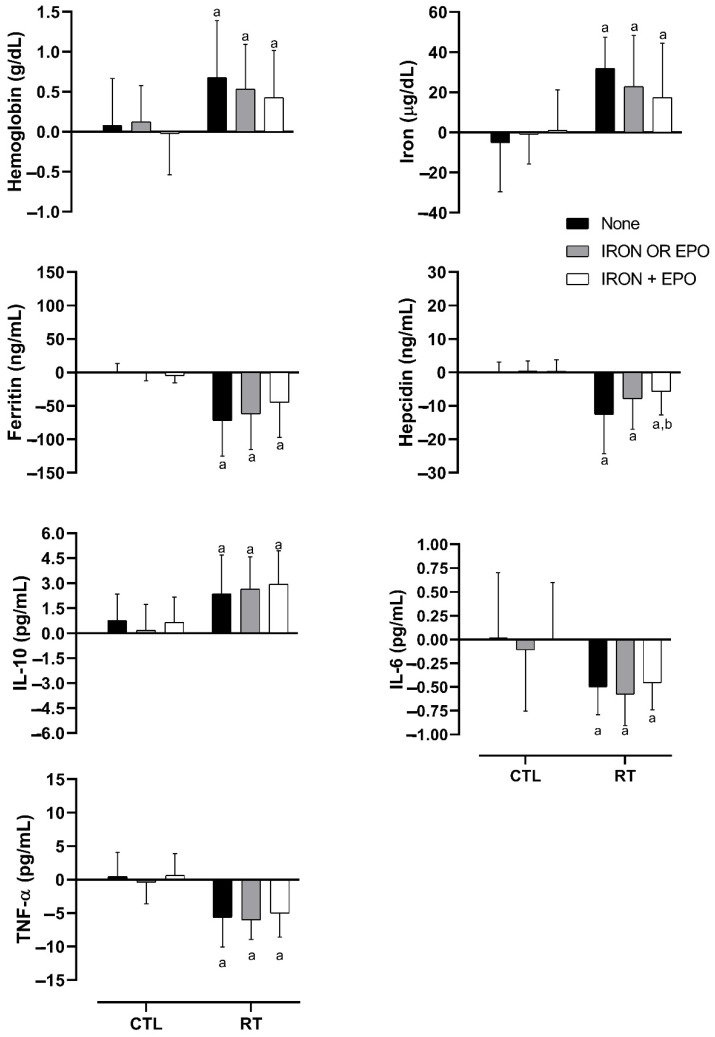
Deltas (post/pre) of hemoglobin, iron, ferritin, hepcidin, and cytokine response to RT following different iron treatments. CTL: control group; RT: resistance training group. ^a^
*p* < 0.05 in relation to the corresponding treatment in CTL. ^b^
*p* < 0.05 in relation to RT—none.

**Table 1 nutrients-13-03250-t001:** Baseline characteristics for each group according to iron treatment.

Variables	CTL			RT			*p* Value	Eta^2^
	None	EPO or IRON	EPO + IRON	None	EPO or IRON	EPO + IRON		
Age (years)	66.16 ± 4	66 ± 4.14	66.58 ± 3.77	66.75 ± 3.34	67.83 ± 3.33	67.28 ± 3.19	0.687	0.005
Body mass (kg)	71.47 ± 14.72	72.33 ± 15.46	73.61 ± 14.38	71.38 ± 16.76	77.03 ± 17.31	73.78 ± 16.41	0.740	0.004
Body mass index (kg/m^2^)	26.54 ± 2.85	26.78 ± 3.12	26.98 ± 2.88	26.66 ± 3.75	27.95 ± 3.86	27.32 ± 3.78	0.769	0.003
Waist circumference (cm)	94.54 ± 12.84	94.91 ± 12.12	96.27 ± 11.87	93.9 ± 12.24	98.12 ± 12.13	95.47 ± 11.41	0.680	0.005

Data expressed as means and standard deviation. CTL: control; RT: resistance training; EPO: erythropoietin.

## Data Availability

Data ara available upon reasonable request to the corresponding author.

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
