# Peer review of "Does the Combined Effect of Resistance Training with EPO and Iron Sulfate Improve Iron Metabolism in Older Individuals with End-Stage Renal Disease?"

_nutrients, 2021, doi:10.3390/nu13093250_

Round 1

Reviewer 1 Report

The authors investigate whether the combined effect of resistance training (RT) with EPO and iron treatment improves iron metabolism in elderly individuals affected by end-stage renal disease. This is an interesting issue to be addressed due to the side effects related to iron and ESA administration. However, I have some major concerns on this work.

Major comments

The authors address two fundamental questions regarding the efficacy of RT alone or in combination with iron/ESA therapy. However, in a previous paper (ref. 6) they have already widely described the beneficial effects of RT alone on iron metabolism. The novelty of this work should be to investigate the combined effect of RT with the conventional therapy while it seems that some important details are missing.

The “Procedures” section in Materials and Methods completely lacks of details on iron/ESA treatment (time and way of administration, doses…) thus it results quite hard to evaluate the efficacy of the treatment on iron metabolism. The authors should revise this section. Moreover, the authors should discuss why CTL patients do not display changes in iron metabolism markers following 6 months of treatment with conventional therapy for ESRD patients. Three times/day for 8 weeks of treatment results in iron level increase in chronic kidney disease patients (ref. 5).

As reported in the discussion by the authors, I suggest to increase the sample size focusing on the RT+ESA that seems the most promising therapy.

Why the authors collect in the same bar graph the data of iron or EPO treatment? They should separate the data to be more informative.   

The authors conclude that “conventional use of ESA+iron in this population is more effective in combination with RT”. Instead, the graphs show that RT alone ameliorates iron metabolism and the combined treatment is equal or even less effective than RT alone. The authors should reconsider the discussion section and the conclusion since they suggest as further studies the subject of the present work.

Minor comments

The authors must revised the abstract: the number of patients enrolled in the different treatment group is not correct and does not match with the scheme in figure 1. Line 25: “RT consisted of…at moderate intensity” of what? The authors should add some more details here or in the “Procedures” section in Materials and Methods or alternatively clearly refer to a previous work.

Figure 2, Hemoglobin graph: the differences in hemoglobin concentration between the treatment groups are really quite small and it’s hard to envisage statistical significance. The authors should check the statistical analysis: is the P< 0.05?

Legend figure 3: “RT improved..” is misleading since RT improves anti- but decreases pro-inflammatory cytokines. Please modify the text accordingly.

Line 129: “improved..” misleading, same comment as above

I think that figure 4 is redundant and not informative since it reports the same data than figure 2/3.

As it concerns the References, five self-citations out of thirteen are too many. The authors should corroborate the discussion with some more citations.

Author Response

Reviewer #1

The authors investigate whether the combined effect of resistance training (RT) with EPO and iron treatment improves iron metabolism in elderly individuals affected by end-stage renal disease. This is an interesting issue to be addressed due to the side effects related to iron and ESA administration. However, I have some major concerns on this work.

Authors response: The authors are grateful for your comments which were truly relevant for the improvement of the manuscript. Thank you.

Major comments

The authors address two fundamental questions regarding the efficacy of RT alone or in combination with iron/ESA therapy. However, in a previous paper (ref. 6) they have already widely described the beneficial effects of RT alone on iron metabolism. The novelty of this work should be to investigate the combined effect of RT with the conventional therapy while it seems that some important details are missing.

Authors Response: Thank you for this comment. We believe that now, after addressing the reviewers’ comments the manuscript is more suitable for publication.

The “Procedures” section in Materials and Methods completely lacks of details on iron/ESA treatment (time and way of administration, doses…) thus it results quite hard to evaluate the efficacy of the treatment on iron metabolism. The authors should revise this section. Moreover, the authors should discuss why CTL patients do not display changes in iron metabolism markers following 6 months of treatment with conventional therapy for ESRD patients. Three times/day for 8 weeks of treatment results in iron level increase in chronic kidney disease patients (ref. 5).

Authors response: Dear reviewer, thank you for this question. The protocol of iron and ESA agents were performed according to the parameters established by the Ministry of Health in Brazil for ESRD. We added a better description and a citation for such treatments [1].  Moreover, we added a paragraph in the discussion section outlining the possibilities of why CTL patients did not displayed changes in iron metabolism.

Find below:

“Methods: Iron and ESA treatments

The protocol of iron and ESA agents were performed according to the parameters established by the Ministry of Health in Brazil for ESRD [1]. Briefly, Patients on hemodialysis are initially treated with one of the following options, later adjusted according to the therapeutic response: -50-100UI/Kg, subcutaneously, divided into 1 to 3 applications per week; 50-100UI/Kg, intravenously, divided into 3 applications per week. If after four weeks of treatment, the hemoglobin elevation is less than 0.3 g/dL per week: increase the dose by 25%, respecting the maximum dose limit, which is 300IU/Kg/week per route subcutaneous and 450 IU/kg/week intravenously. If after four weeks of treatment, the hemoglobin elevation is in the range of 0.3 - 0.5 g/dL per week: keep the dose in use. If after four weeks, the hemoglobin elevation is greater than 0.5 g/dL per week or the hemoglobin level is between 12 and 13g/dL: reduce the dose by 25% to 50%, respecting the minimum dose limit recommended, which is 50UI/Kg/week subcutaneously. Temporarily suspend treatment if the hemoglobin level is above 13 g/dL.

Treatment should be continuously, targeting the hemoglobin at 11 g/dL. Temporary interruption of treatment is recommended if the hemoglobin level is above 13 g/dL, with resumption when hemoglobin levels are below 11 g/dL. Discontinuation should be considered in the event of a serious adverse event resulted from it.

Iron III hydroxide saccharate is for intravenous use and is presented in 5ml ampoules containing 100mg of iron III (20mg/ml). It must be diluted in 100ml of saline solution and infused within 15 minutes, according to the manufacturer. A study demonstrates its safe use in shorter administration times, up to 5 minutes, without increasing adverse reactions.” (Line 82)

“Discussion:  According to the results found by Agarwal et al. [2], non-hemodialysis CKD patients on oral iron therapy have improved hemoglobin, TIBC, transferrin saturation and ferritin. However, although our participants also received oral iron supplementation, they were also on hemodialysis treatment, which is known to increase iron losses. It is estimated that CKD patients on hemodialysis lose 1 to 3 g of iron per year, which coupled with the fact that hemodialysis patients have particularly impaired dietary iron absorption appears to make oral iron supplementation poorly effective in improving iron markers [3]. In fact, oral iron supplementation was no better than placebo in improving anemia, improving, or preventing iron deficiency, or reducing ESA dosage in hemodialysis patients [3-6].  Therefore, the findings from the present study can provide clues on the application of RT in this population to counteract iron-related diseases. To date, no studies have investigated the pooled effects of exercise plus ESA therapy or iron supplementation after 24 weeks.” (Line 183)

As reported in the discussion by the authors, I suggest to increase the sample size focusing on the RT+ESA that seems the most promising therapy.

Authors response: The authors agree with this commentary. As we couldn’t increase our sample size, we add in the discussion the importance of doing that in further studies. Thank you.

Why the authors collect in the same bar graph the data of iron or EPO treatment? They should separate the data to be more informative.   

Authors response: The authors agree with the reviewer. The initial idea was to separate one group for iron and another group for EPO treatment. However, this would reduce the sample size of each group increasing the risk to occur statistical errors. In this case, the most adequate was to pool them as a single group.  

The authors conclude that “conventional use of ESA+iron in this population is more effective in combination with RT”. Instead, the graphs show that RT alone ameliorates iron metabolism and the combined treatment is equal or even less effective than RT alone. The authors should reconsider the discussion section and the conclusion since they suggest as further studies the subject of the present work.

Authors response: Thank you for this commentary. We emphasize the effect of RT alone in ameliorating iron metabolism in both discussion and conclusion. (Line 224-225) 

Minor comments

The authors must revised the abstract: the number of patients enrolled in the different treatment group is not correct and does not match with the scheme in figure 1. Line 25: “RT consisted of…at moderate intensity” of what? The authors should add some more details here or in the “Procedures” section in Materials and Methods or alternatively clearly refer to a previous work.

Authors response: Thank you for noticing that. We revised the abstract and added that the RT consisted of 24 weeks/3 days per week at moderate intensity of full-body resistance exercises. Moreover, in the procedures section in Materials and Methods we clearly refer to the previous work. Thank you for this suggestion.

The protocol of RT occurred before the hemodialysis section and was described in detail by Moura et al.” (Line 79)

Figure 2, Hemoglobin graph: the differences in hemoglobin concentration between the treatment groups are really quite small and it’s hard to envisage statistical significance. The authors should check the statistical analysis: is the P< 0.05?

Authors response: We checked the statistical analysis. The P was < 0.05 for this variable. Find below the detailed P value.

Hemoglobin

P value

None:RT pre vs. None:RT post

<0,0001

None:RT pre vs. EPO or IRON:RT post

0,0154

None:RT pre vs. EPO + IRON:RT post

0,0035

EPO or IRON:RT pre vs. EPO or IRON:RT post

0,0061

EPO + IRON:CTL pre vs. EPO + IRON:RT post

0,0378

EPO + IRON:CTL post vs. EPO + IRON:RT post

0,0297

EPO + IRON:RT pre vs. EPO + IRON:RT post

0,0001

Legend figure 3: “RT improved..” is misleading since RT improves anti- but decreases pro-inflammatory cytokines. Please modify the text accordingly.

Authors response: We corrected the legend of figure 3 we changed to “cytokines regulation following RT in hemodialysis patients”

Line 129: “improved..” misleading, same comment as above

Authors response: improved was removed from these sections.

I think that figure 4 is redundant and not informative since it reports the same data than figure 2/3.

Authors response: Dear reviewer, thank you for this suggestion. We would like to ask you if we can keep the figure 4 within the manuscript once delta values might help readers in understanding how the data respond to the treatments.

As it concerns the References, five self-citations out of thirteen are too many. The authors should corroborate the discussion with some more citations.

Authors response: We added more references since we discussed more about the phenomenon after the reviewer comment. Thank you.

Reviewer 2 Report

The authors conduct a study that known effect of resistance training (RT) reduced the blood level of pro-inflammatory cytokine (TNF-alpha and IL-6) and elevated the anti- inflammatory cytokine (IL-10); resulted in lower hepcidin and improving iron utilization for older individuals with end-stage renal disease. It is interesting and practical to aid the patients care. Some questions need to clarify. 1. These patients had good hemoglobin initially, they need few EPO and iron in the routine care. Is the benefit persistent for anemic patients? 2. Those RT patients were slightly obese (compared to CTL and normal populations as well), it represents mild low risk of inflammation status? 3. Other biochemical data including the nutritional marker and electrolytes were absent. It carry important clue for practice the resistance training. 4. There was no detailed descriptions about resistance exercise performed in the text, intra-dialytic? And equipment supplied? 5. C-reactive protein (CRP) is a more practical method for routinely care of dialysis patients. Was there involved in this study? 6. Several factors also influence the inflammation; for example, Vit C, larger EPO dose… were not introduced during the period of the study. If authors can point out which character of dialysis patients will obtain more benefit from the exercise. Its impact is profound in ordinary care.

Author Response

Reviewer #2

The authors conduct a study that known effect of resistance training (RT) reduced the blood level of pro-inflammatory cytokine (TNF-alpha and IL-6) and elevated the anti- inflammatory cytokine (IL-10); resulted in lower hepcidin and improving iron utilization for older individuals with end-stage renal disease. It is interesting and practical to aid the patients care. Some questions need to clarify.

Authors response: The authors would like to thank for the reviewer kind comments. We are sure that, after addressing all the suggestion the manuscript improved in quality and content. Thank you.

  1. These patients had good hemoglobin initially, they need few EPO and iron in the routine care. Is the benefit persistent for anemic patients?

Authors response: Truly relevant this question. Based in the present literature, it is rationale to infer that patients with anemia can be beneficiated more than patients with good hemoglobin.  

  1. Those RT patients were slightly obese (compared to CTL and normal populations as well), it represents mild low risk of inflammation status?

Authors response: Obesity is an inflammatory condition with an increased pro-inflammatory cytokine and decreased anti-inflammatory proteins. However, considering that the inflammatory proteins did not differ from baseline in any group. Those RT slightly obese compared to CTL appear to not influence in the inflammatory status in the present study.

  1. Other biochemical data including the nutritional marker and electrolytes were absent. It carry important clue for practice the resistance training.

Authors comments: Thank you for this suggestion. Indeed, the lack of nutritional markers and electrolytes may limit our data. We added a statement about that in the limitations of the study. (Line 222-223)

  1. There was no detailed descriptions about resistance exercise performed in the text, intra-dialytic? And equipment supplied?

Authors response: We added in the methods section the citation of the work that used the same resistance training protocol. (Line 79)

  1. C-reactive protein (CRP) is a more practical method for routinely care of dialysis patients. Was there involved in this study?

Authors response: Thank you for this comment, you are correct about the CRP being a more practical method for routinely care of dialysis patients. Although we did not assess this protein, CRP is a product of the IL-6 and TNF signaling. These two cytokines induce the increase of the production of acute phase proteins in the liver, including CRP. Therefore, the rationale to analyze IL-6 and TNF instead of CRP is to verify if the principle of the mechanisms of inflammation changed following exercise.

  1. Several factors also influence the inflammation; for example, Vit C, larger EPO dose… were not introduced during the period of the study. If authors can point out which character of dialysis patients will obtain more benefit from the exercise. Its impact is profound in ordinary care.

Authors response: Truly relevant this comment, thank you for this perspective. We added in the discussion section a better description of the impact of our study in the ordinary care. Moreover, we added in the methods section the dose of EPO and iron. Finally, the main message of the study is that regardless of the treatment, resistance training might be an important tool for a better prognostic of this population.

Conclusion: “Therefore, it is rationale to infer that the application of RT programs should strongly be recommended in dialysis care. This would improve the prognosis of several ESRD patients, especially those with iron deficiency and anemia.” (Line 233-236)

Round 2

Reviewer 1 Report

The authors have improved the Materials and Methods together with the Discussion section and they have convincingly replied to many of my comments.